# Qualitative study evaluating the expectations and experiences of Dutch parents of children with chronic gastrointestinal symptoms visiting their general practitioner

Sophie M Ansems ,[1] Ilse N Ganzevoort ,[1] Donald G van Tol,[1,2] Tryntsje Fokkema,[1] Marijke Olthof,[1] Marjolein Y Berger,[1] Gea A Holtman [1]

¹Department of Primary and Long-term Care, University Medical Center Groningen, Groningen, The Netherlands
²Department of Sociology, Faculty of Behavioral and Social Sciences, University Medical Center Groningen, Groningen, The Netherlands

**Correspondence to**
Sophie M Ansems;
s.m.ansems@umcg.nl

## ABSTRACT

**Objectives** Chronic gastrointestinal symptoms are common among children and affect their daily activities and quality of life. The majority will be diagnosed with a functional gastrointestinal disorder. Effective reassurance and education are, therefore, key components of the physician's management. Qualitative studies have shown how parents and children experience specialist paediatric care, yet less is known about general practitioners (GPs), who manage most cases in the Netherlands and have a more personal and enduring relationship with their patients. Therefore, this study evaluates the expectations and experiences of parents of children visiting a GP for chronic gastrointestinal symptoms.

**Design** We conducted a qualitative interview study. Online interviews were audio and video recorded, transcribed verbatim and independently analysed by the first two authors. Data were collected and analysed concurrently until data saturation was reached. Using thematic analysis, we developed a conceptual framework reflecting respondent expectations and experiences. We performed a member check of the interview synopsis and the conceptual framework.

**Setting** Dutch primary care.

**Participants** We purposively sampled participants from a randomised controlled trial evaluating the effectiveness of faecal calprotectin testing in children with chronic gastrointestinal complaints in primary care. Thirteen parents and two children participated.

**Results** Three key themes emerged: disease burden, GP–patient relationship and reassurance. Often, the experienced disease burden and the pre-existing GP–patient relationship influenced expectations (eg, for further investigations or a sympathetic ear), and when a GP fulfilled these expectations, a trusting GP–patient relationship ensued that facilitated reassurance. We found that individual needs influenced these themes and their interrelationships.

**Conclusion** Insights provided by this framework could help GPs managing children with chronic gastrointestinal symptoms in daily practice and may therewith improve the consultation experience for parents. Further research should evaluate whether this framework also holds true for children.

## STRENGTHS AND LIMITATIONS OF THIS STUDY

⇒ This is the first study to specifically address parental expectations of and experiences with a GP visit for paediatric chronic gastrointestinal symptoms.
⇒ Coding and analysis were done by several researchers and two separate member checks were performed, increasing the study's credibility.
⇒ Although our intention was to study the expectations and experiences of children as well, only two children participated in this study.
⇒ Despite purposive sampling, all participants had a typical Dutch cultural background.

**Trial registration number** NL7690.

## INTRODUCTION

Chronic gastrointestinal (GI) symptoms are prevalent in school-aged children, ranging from 0.3% to 19% in Western countries.[1] In the Netherlands, a general practitioner (GP) will typically see 10 children with chronic GI symptoms each year,[2 3] of whom about nine will have a functional GI disorder (FGID).[3 4] FGID (eg, functional constipation) is one of the symptom clusters of 'functional somatic symptoms' (FSS, ie, symptoms that cannot be fully explained by well-defined psychiatric or somatic illness).[4 5] Other FSS clusters in children are pain, fatigue and cardiopulmonary symptoms.[5] In these settings, GPs can reassure and educate parents and children through various strategies, including normalisation,[6] explanation,[7 8] affective reassurance,[9] the exclusion of physical illness,[7 10] and the provision of reassurance through normal test results.[11–13] However, research has shown that 50% of children with FGID still have abdominal pain that affects daily activities 12 months

**Table 1** Parents' and children's characteristics (n=15)

| Characteristic | Number |
|---|---|
| Gender child, male * | 7 |
| Age child, years * | |
| 4–7 | 5 |
| 8–11 | 8 |
| 12–15 | 1 |
| 16–17 | 1 |
| Urban area of residence * | 5 |
| Country of birth of both parents † | |
| The Netherlands | 15 |
| Educational level mother‡ † | |
| Low | 1 |
| Intermediate | 6 |
| High | 8 |
| Educational level father†‡ | |
| Low | 2 |
| Intermediate | 6 |
| High | 7 |
| Paid employment † | |
| Both parents | 14 |
| One parent | 1 |
| Marital status parents † | |
| Married / living together | 11 |
| Divorced | 4 |
| Present at interview † | |
| Child | 1 |
| Child and mother | 1 |
| Mother | 8 |
| Father | 2 |
| Both parents | 3 |
| Diagnosis after 6 months* | |
| Functional constipation | 9 |
| Irritable bowel syndrome | 2 |
| Functional constipation and functional abdominal pain | 2 |
| Functional abdominal pain | 2 |

*Clinician reported
†Patient reported
‡Educational level was divided in low (primary and lower secondary education), intermediate (secondary (vocational) education) and high (Bachelor's degree or higher).[46 47]

after first presenting,[14] while other research has shown an association between FGID and the child's quality of life, school absenteeism, anxiety and depression.[15 16]

Most current reassurance and education strategies are planned from the GP's perspective, with evidence that GPs do not always assess patient expectations and needs adequately.[13] Qualitative studies among children and adolescents with FGID revealed their desire of a clear diagnosis provided by their treating physician.[17–19] However,

qualitative studies performed in paediatric specialist care have shown that pre-existing concerns (eg, about serious illness or the impact of symptoms) and expectations of the doctor (eg, to exclude physical illness or help manage pain) varied considerably among parents,[20–23] creating uncertainty about the best reassurance strategy.[24] Parents may also hold different views about the treating doctor, perhaps believing another professional would be better suited to treat their child or that the doctor does not consider psychologically based symptoms to be genuine.[20] Other parents report that they want guidance and follow-up by their usual GP or medical professional.[22] Indeed, most children with chronic GI symptoms are managed in primary care in the Netherlands,[3] yet most research has taken place in paediatric specialist care.[17–23] We hypothesised that the more personal and enduring relationship between GPs and patients could influence how parents and children experience consultations. In this study, we aimed to evaluate the expectations and experiences of parents and children when visiting a GP with chronic GI symptoms through a qualitative interview study.

## METHODS
### Study design
We used an inductive–deductive methodology,[25] meaning we used both pre-existing insights and literature as well as new insights that arose during the analysis of our data. We included parents and children living in the northern part of the Netherlands and collected data between January and June 2021. To allow personal and sensitive themes to arise, we collected data via in-depth semi-structured interviews.[26] We used the Consolidated Criteria for Reporting Qualitative Research checklist[26] and the Standard Reporting of Qualitative Research checklist.[26 27]

### Respondents
Parents and children engaged in an ongoing randomised controlled trial (RCT) evaluating the clinical and cost effectiveness of faecal calprotectin testing in primary care were eligible to take part[28] (The Netherlands Trial Register: NL7690). The trial included children aged 4–18 years who visited their GP with chronic abdominal pain and/or chronic diarrhoea and excluded children with a history of chronic organic GI disease or who had received endoscopic evaluation, paediatric referral for GI symptoms or faecal calprotectin testing within the preceding 6 months. The first author approached parents and/or children by telephone after receiving informed consent for their participation in the RCT and related research.

For the present study, we used purposive sampling by age, sex (children and parents), parental educational level and parental country of birth. Inclusion continued until data saturation, defined as being when interviews generated no new information on the research questions. All respondents received an email with a patient information letter and provided digital informed consent, which

**Table 2** Subthemes per theme and factors positively and negatively influencing each theme

| | Subtheme | Positive influence | Negative influence |
|---|---|---|---|
| Disease burden | Symptoms | Mild symptoms | Long duration of symptoms |
| | | Stable symptoms | Recurring symptoms |
| | | Short symptom duration | Additional symptoms (eg, weight loss) |
| | Impact | No limitations due to symptoms | Symptoms impact rest of family |
| | | – | Child is limited in everyday life due to symptoms (eg, school absenteeism and sports) |
| | Concerns | No concerns about symptoms | Concerns about symptoms |
| | | No concerns because good explanation for symptoms (eg, constipation, or runs in family) | Concerns about additional symptoms (eg, weight loss) |
| | | – | Concerns about impact on everyday life |
| | History | – | Parents recognise symptoms from their own medical history |
| | | – | Other healthcare providers could not relieve the symptoms (eg, psychotherapist, complementary medicine) |
| Doctor–patient relationship | Pre-existing relationship | Long-lasting doctor–patient relationship | First time at this GP |
| | | Positive previous experience with GP | Negative previous experience with GP for chronic abdominal symptoms |
| | Consultation | Parents felt heard and understood | Not enough room in consultation for respondent's story |
| | | GP focused on and comforted child | Short duration of consultation |
| | | GP showed personal interest | GP did not answer main request |
| | | GP fulfilled expectations | GP normalised or downplayed symptoms |
| | | GP and respondents together determined treatment plan | GP showed no empathy |
| | Follow-up | GP planned follow-up consultation | No planned follow-up consultation |
| | | Low threshold to make new GP appointment | Parent had to contact GP about continuation |
| | | – | GP did not take enough initiative |
| | | – | GP provided no help besides medical care (eg, psychosocial care) |
| | | – | GP did not provide a solution |
| Reassurance | Explanation | Matching explanatory model | Non-matching explanatory model |
| | | Explanatory model fulfilled the respondent's expectation (eg, constipation) | GP provided unclear or no explanatory model |
| | | Respondents and GP together explored different explanatory models | GP did not seem to believe in explanatory model |
| | | – | Respondents were unsure about explanation |
| | Physical interventions | Successful treatment (eg, Forlax relieved symptoms) | Unsuccessful treatment |
| | | Physical examination confirmed diagnosis | Medical tests did not reveal an explanation |
| | | | Parents wished further medical investigation |
| | | – | GP provided 'quick fix' for long-lasting problem (eg, medication) |

GP, general practitioner.

consistent with Dutch law, was obtained either from parents (child age <12 years), parents and child (age 12–15 years) or the child only (age >15 years).

**Data collection**

An interview guide was prepared using sensitising concepts from the literature and expert discussion.

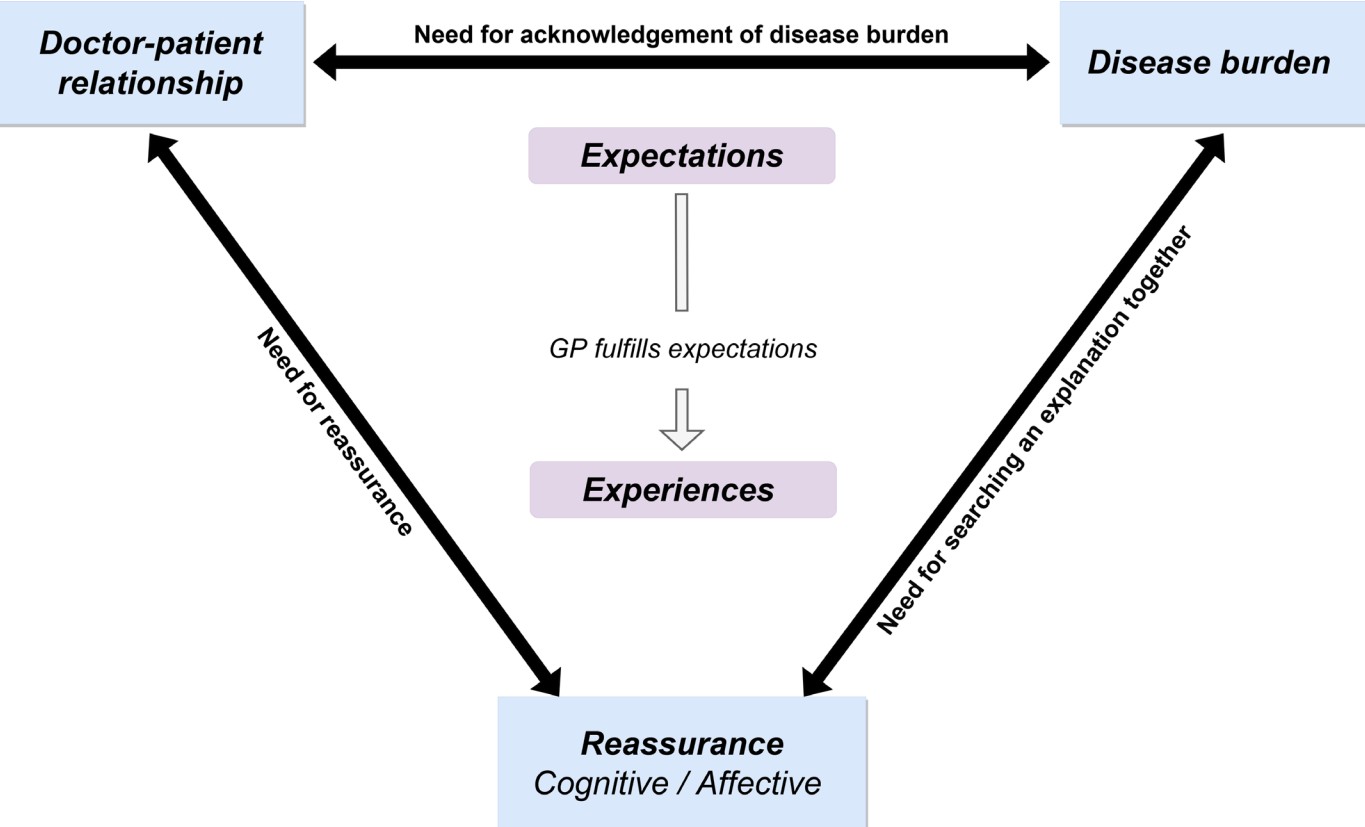

**Figure 1** **Conceptual framework**. All three themes (blue rectangles) influenced the expectations and experiences of parents, and their experiences were influenced by whether the GP met their expectations. GPs can influence the relationships between themes by responding to the needs of parents and children (displayed in BOLD, next to the arrows).

Sensitising concepts comprised reassurance and education strategies, parental worry cognitions, parental and child expectations, views about the GP, and experiences of the GP consultation.[6–9 11–13 20–24] The interview guide contained open-ended questions concerning the incentives, expectations and experiences related to the visit, as well as the GP's explanation for the symptoms (online supplemental file 1).

The first author, a female physician and researcher trained to perform qualitative interviews, conducted the semi-structured interviews. In the interview setting, she presented herself as a researcher and not a physician. By doing so, she prevented confusion about her role and gave the respondents the feeling they could speak freely about their GP.[29 30] Another primary care researcher who is not a physician (second author, a PhD candidate studying FGID in children) observed all interviews. Before starting, we emphasised the confidential nature of the interview, that only anonymised transcripts would be kept, deleting any recordings and that data would not be discussed with their GP. Children aged ≥12 years could decide if a parent accompanied them, but we only interviewed the parents for children aged <12 years. Interviews were held online (via Microsoft Teams) due to the COVID-19-pandemic and at a time convenient for respondents. They lasted 15–45 min, were audio and video recorded, and transcribed verbatim by the interviewer. Quotes were translated from Dutch to English by the first author and subsequently edited by a native English speaker and editor. Finally, the first author checked whether their meaning retained. Parents and/or children received an interview synopsis and checked whether the interpretation of the interview was correct (member check 1).

**Analysis**
Data were collected and analysed concurrently, allowing emerging themes to be incorporated and explored in subsequent interviews. Thematic analysis was conducted as proposed by Braun and Clarke.[31] The first and second author coded the first three transcripts separately for standardisation. The first author then analysed the remaining transcripts and the second author independently checked the coding results before they discussed and resolved inconsistencies by consensus. Finally, two other members of the research team (a sociologist and a primary care researcher) checked the coding tree. Any remaining inconsistencies and missing codes were discussed during a research team meeting.

Throughout the analysis, memos were written, including notes about the identified themes and their relationships, and provisional findings were critically discussed in research team meetings. We developed an initial conceptual framework after four meetings, following which the first author checked and confirmed whether it could

be applied to each respondent's story. Further member checking was undertaken by inviting three respondents to comment on their own story. All analyses were facilitated by the use of Atlas.ti V.9.1 software.

## Patient and public involvement

Parents and children were involved in the interviews and its interpretation. Additionally, the results were disseminated among them, as will the manuscript once published.

## RESULTS
## Respondents

We approached 23 eligible participants by telephone, of which 18 agreed to receive further information. Two parents and one child declined to participate either for unknown reasons, personal circumstances or not wanting to talk (one each). Data saturation first appeared after 10 interviews when we had only interviewed 1 child aged ≥12 years and 1 father. Therefore, we continued purposive recruitment of these groups for an additional five interviews to improve representation and saturation. Unfortunately, only 1 more child was recruited. All interviews took place within 2 months after the initial presentation. Table 1 shows the characteristics of the 15 respondents. The results of the initial member check for these 15 respondents are shown in online supplemental file 2 (member check 1), detailing that only one mother returned the synopsis with clarifying information.

## Themes

We discovered three themes concerning the expectations and experiences of parents when visiting a GP for chronic GI symptoms: disease burden, GP–patient relationship and reassurance, with specific needs affecting their interrelationship. Using illustrative quotations, we discuss each theme's meaning, subthemes (table 2) and relation to the expectations and experiences of respondents, seeking to reveal the relationship between themes reinforced by respondent need (conceptual framework in figure 1). Three cases were purposively selected for additional member check (online supplemental file 2, member check 2), as detailed in figure 2A–2C as representative examples.

## Disease burden

The disease burden experienced by parents influenced their expectations of GPs. Burden comprised the duration and severity of symptoms, the presence of additional symptoms (eg, weight loss), the impact on everyday life and the degree of concern about the symptoms (table 2). Some parents experienced low disease burden (eg, when in an early stage of symptoms), while others experienced chronic recurring GI symptoms that had limited everyday life for years.

Mother (M8): 'For one year he was pain free and … during the last couple of months it returned. He has lost … weight, and the pain is limiting him in daily

life … and in school. He arrived late to school because his belly hurt and he had to go to the toilet again. So, I just wanted further investigation.'

Most parents with a low disease burden emphasised that they had no specific expectations of the GP.

Mother (M6): 'Yes, exactly [the GP fulfilled my expectations]. Because it was not a severe situation, I did not have high expectations. I did not have a tense feeling,… so… GPs can help just by being friendly.'

Parents had higher expectations with higher disease burdens, typically expressing concern that 'something must be wrong'. This often resulted in questions about the cause of symptoms and expectations for the GP to label symptoms, conduct further investigations, or refer to secondary care.

Father (F1): 'At some point you are fed up… you want to know the answer, where it could come from? I know that my symptoms are clearly due to stress and everything, but I do not think that is the case for [my daughter]."

Mother (M3): 'And then I said 'You did not find anything but I would like it to be further investigated because 1) she is losing weight and 2) she just has an awful lot of pain attacks while turning pale, feeling bad, and not eating'. Her stool is almost never normal, so that cannot be right. But then, I got the response 'What investigations were you thinking of?'… I am no doctor! So… I said… 'Just send me to a paediatrician'… 'There must be a cause for this, this cannot be right'.'

Concern about symptoms varied among the parents. A couple of parents expressed concern over additional symptoms (eg, fever), while others worried about the impact of symptoms on everyday life (eg, school absenteeism and disturbed sleep) or the lack of explanation. By contrast, a few parents emphasised that they usually do not worry easily about their child's health and did not in this case.

Although both children mentioned that their symptoms affected them negatively during school and that they hoped the GP could find the cause and a solution, their parents had taken the initiative to consult the GP in both cases.

## Doctor–patient relationship

The doctor–patient relationship comprised two elements: the pre-existing relationship (influencing both expectations and experiences) and the relationship during and after the consultation (influencing only experiences). Most parents reported a trusting relationship with their GP, with several having long-standing relationships and good previous experiences. One father expressed that his GP '*is not only a doctor, but above all, a good human being*'.

Parents emphasised the importance of feeling heard and understood by the GP during consultations.

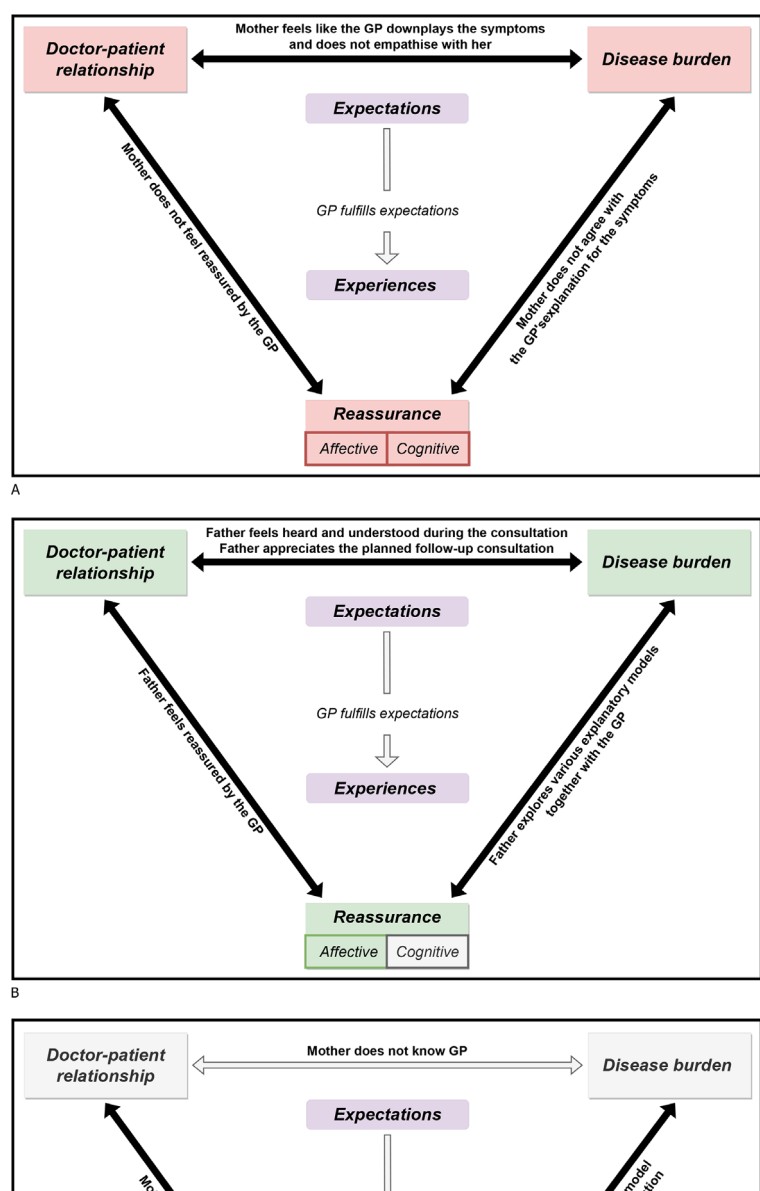
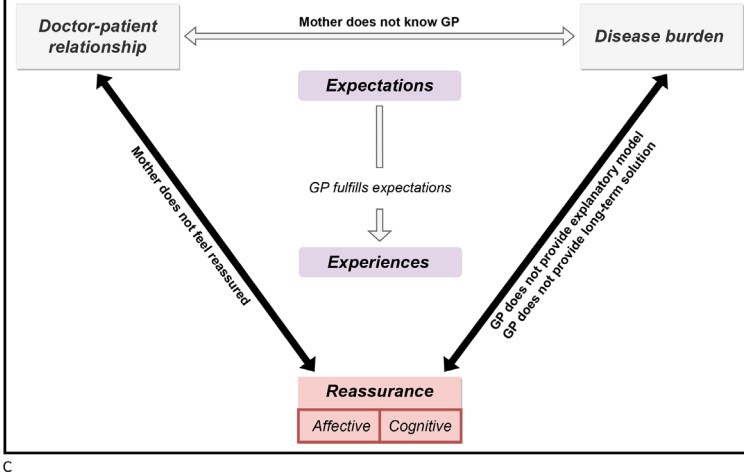

**Figure 2** (A) Conceptual framework exemplified by Case 8. All themes are shown in red, indicating that these were negatively influenced, because the GP did not respond to the mother's needs. The three bold arrows indicate a strong relationship between the themes. (B). Conceptual framework exemplified by Case 12. The themes shown in green were positively influenced because the GP responded to the father's needs. The theme shown in grey (cognitive reassurance) was not prominent during the interview. Bold arrows indicate the strong relationship between themes. (C). Conceptual framework exemplified by Case 9. The red theme was negatively influenced because the GP did not respond to mother's needs. The grey themes were not prominent during the interview. The bold arrows indicate strong relationships between themes, and the grey arrow indicates that this relationship was not prominent during the interview.

Additionally, most appreciated the GP comforting the child and directing his/her questions and explanations to the child.

Mother (M4): 'If something is wrong … he just really takes the time for you. Sometimes I am in the waiting room and the previous consultation is running late. But, actually, I find that a good sign because it means he takes the time for you. Also, for children, he really takes the time.'

Others felt their GP downplayed the child's symptoms.

Mother (M8): 'But I do find that he could have shown a bit more empathy and … talked with my child a bit more. Because what he is feeling … is really what he is feeling. And now he has the sense that he was pretending. But it actually is … real.'

Parents also felt comforted when the GP planned a follow-up consultation. This made them feel that the GP had acknowledged their concerns and the disease burden.

Mother (M2): 'She completely examined [my child], we had a good conversation, and she [said] we will see and to call in 2 weeks to discuss it again… it was not like… manage it yourself and goodbye. Not at all.'

Similarly, parents felt isolated when the GP did not plan a follow-up consultation.

Mother (M1): 'If everything is ruled out [and] there are still symptoms, as a parent you want to do something… I would have appreciated a follow-up appointment, preferably already scheduled, so you can talk about it to find a good solution together. Although he cannot do anything with the test result [everything is ruled out]… let's keep talking… like… what can I help you with? Do you actually need help, or do you think you can handle it on your own?'

When younger, the 17-year-old respondent mentioned that he found his GP intimidating. However, recent consultations for his chronic GI symptoms had changed this because the GP was considered '*friendly and neat*' towards him.

### Reassurance

Parents expressed to feel reassured via two different pathways; cognitive and affective reassurance. Cognitive reassurance was achieved when the GP responded to or changed their beliefs about the symptoms (their cognitions). Whether the GP provided an explanatory model for the symptoms, and the choice of model, played prominent roles in achieving cognitive reassurance. Sometimes the GP gave an explanation that matched the parents' belief system (eg, the GP confirmed their expectation that constipation caused the symptoms), making it relatively easy to establish cognitive reassurance. However, if their child's symptoms did not conform to the explanatory

model, parents experienced insecurity about the explanation for the symptoms.

Father (F5): 'That is the weird part because you know that the laxatives are for the constipation, but if a child suddenly has a fever and lies on the couch while having a hard time breathing… and has recovered within one hour… you do not think it is constipation … but … we did not know whether to call the GP again.'

How the GP conveyed the explanatory model also had a considerable impact on cognitive reassurance.

Child (Ch7): 'I don't know … it did feel like a bit of an oversimplification. Like … it felt like she did not know the answer herself… [so] she called it irritable bowel… that is how it felt.'

This was especially the case for the two children, who both expressed that they did not understand the explanations because 'the GP talked too much' and 'used too many difficult words'.

Some parents emphasised that positive findings during physical examination or successful treatment made them believe in a certain explanatory model.

Mother (M4): 'Yes, I think [the GP's final diagnosis was heartburn], although this was not said definitively … [but] since we put the bed up based on their advice,… I assume that it is related to heartburn.'

Other parents expressed that they wanted the GP to perform more medical tests to find an explanatory model for the symptoms. However, none of the parents mentioned to feel reassured by a normal test result.

Not all parents wanted a specific explanatory model for the symptoms. Several stated that they just followed the GP's instructions (eg, to take laxatives and return after 2 weeks with an update) and were comfortable that the GP had not provided a specific explanatory model early in the diagnostic process. In these cases, affective reassurance played a larger role, meaning parents felt reassured because the GP gave them the feeling there was no need to worry and they felt heard and understood by the GP.

Father (F14): 'She took both questions seriously … as well as our suggestions … and she found those suggestions plausible. So, she first performed a physical examination, looked at his tummy, knocked on it and she concluded that it was unlikely a purely physical cause.'

### Relationship between the themes and needs of parents and children

The three themes not only influenced the expectations and experiences of parents but also influenced each other (figure 1), with this differing between interviews (see figure 2A–2C). For example, the mother in figure 2A suffered a high disease burden, which prompted her to seek an explanation (cognitive reassurance). In figure 2B,

the strong and long-term doctor–patient relationship made the father trust his GP when told that he should not worry about his son's symptoms (affective reassurance), without a need for an explanation for the symptoms or treatment (cognitive reassurance). By contrast, the doctor–patient relationship in figure 2C was weak, resulting in a greater need for cognitive reassurance because the GP could not offer affective reassurance.

The relationship between the themes was also reinforced by whether the GP fulfilled or neglected the needs of parents. For example, in figure 2B, the GP fulfilled the father's need for acknowledgement of the disease burden by planning a follow-up consultation. This, in turn, decreased the disease burden. Conversely, in figure 2A, the GP downplayed the symptoms and disregarded the need for an explanation, increasing the disease burden and need for cognitive reassurance, respectively.

## DISCUSSION

This qualitative interview study of the expectations and experiences when visiting a GP for chronic GI symptoms revealed three key themes: disease burden, GP–patient relationship and reassurance. The basis of our conceptual framework is that the disease burden and pre-existing GP–patient relationship influence expectations. When a GP met parental expectations, a trusting doctor–patient relationship ensued, creating an environment that allowed affective and cognitive reassurance. Within this framework, specific needs of parents influenced the themes and their interrelationships: for the GP to acknowledge the disease burden, to search for an explanation together with the GP and to feel reassured. GPs should be aware that when they recognise and respond to these needs, they can positively influence the consultation experience of parents.

In this study, parents felt reassured through different pathways, with the success of a given reassurance strategy being situation specific. Sometimes a warm GP–patient relationship, instilling feelings of safety and comfort, was sufficient to provide reassurance (affective reassurance). Other times, parents needed cognitive reassurance, for example through an explanatory model for the symptoms. The need for an explanatory model is in concordance with a Norwegian study among parents of children with chronic abdominal pain.[21] It is already known that specific concerns and beliefs about their symptoms (worrying cognitions) largely determine a patient's experience of cognitive reassurance.[24] While some studies have argued that affective reassurance is a prerequisite to achieving cognitive reassurance,[7] others state that affective reassurance only has a short-term and transient effect.[9] Our findings suggest that any reassurance strategy is situation specific, being related to both the GP–patient relationship and the disease burden.

Our findings fit with the theory of contextualising care, which advocates for medical decisions to be based on both biomedical evidence and the individual patient's context.[32] This study found that important contextual factors, such as emotional state (eg, parental concern) and the GP—patient relationship, affected parental expectations and experiences. Likewise, pre-existing beliefs about symptoms influenced parental adoption of the GP's explanatory model. Some parents felt like their GP did not pay enough attention to contextual factors, resulting in 'contextual errors' (figure 2A),[32] adding to the body of evidence that contextual factors should be used in medical decision-making,[32 33] as recommended by Dutch GP guidelines.[34 35] Research has shown that this is especially important in patients with FSS.[36] Since all children in our study were diagnosed with FGID, a symptom cluster of FSS, our study can be taken to endorse the use of contextual factors in these patients.

The Dutch GP guideline advises that practitioners schedule a follow-up consultation for children with chronic abdominal pain.[34] Our findings support the importance of these follow-up consultations to parents and children, not only making them feel that the GP acknowledged their disease burden but also strengthening their relationship. Hence, a planned follow-up consultation is an important step in creating an enduring doctor–patient relationship, which is a core principle of general practice that has various positive effects on both patient health and healthcare costs.[36–39]

To the best of our knowledge, there is no other study in which parents and children were interviewed about their expectations and experiences of a GP consultation when the child has chronic GI symptoms. This is important because, in a gatekeeping system such as that in the Netherlands, the GP is the first physician to see a child with chronic GI symptoms and will typically manage most cases without referral to secondary care. Therefore, our results are transferable to other countries with gatekeeping systems, such as the UK and Canada.[40] Additionally, the detailed description of our findings makes them transferable to the parents of children with other FSS (eg, chronic fatigue) in general practice, because FSS share a similar underlying multifactorial pathophysiology.[41]

This study knows some limitations. Despite the intention to interview children, only 2 children aged ≥12 years participated. While their expectations and experiences fit within the conceptual framework, they differed from the parent perspective. More importantly, the small number of participating children does not allow us to draw conclusions about their expectations and experiences. This implies that our findings are only transferable to parental expectations and experiences. Future studies should explore children's perspectives, evaluate the framework's applicability to their expectations and experiences. Additionally, all participants had a typical Dutch cultural background. Almost a quarter of the Dutch population are first- or second generation immigrants, and each could have different expectations and experiences of healthcare that vary with their cultural beliefs.[19 42 43] Although we purposively sampled mothers and fathers, we did not find any difference between them.

The COVID-19 pandemic forced us to conduct video-interviews, which may have reduced credibility since in-person interviews are considered the 'gold-standard' for qualitative research. However, recent studies indicate that both methods produce similar data volumes, topic breadths and personal information sharing.[44 45] Additionally, video interviews are more popular among respondents due to scheduling ease, less intimate environment and lack of travel.[45]

A strength of this study is that all respondents were asked to provide feedback on a synopsis of their interview (member check 1) and the three respondents who constituted the representative cases were asked to provide more detailed feedback about our interpretation of their stories (member check 2). Another strength is that two researchers with different backgrounds performed all the interviews and main analyses, while another five researchers (two GPs, a primary care researcher, epidemiologist and a medical sociologist) participated in the coding, thereby offering different theory perspectives.

In conclusion, our findings suggest that an interrelationship exists between disease burden, the GP–patient relationship, and the reassurance given to parents and children who visit a GP for chronic GI symptoms. Although in our study the appropriate GP response was situation specific, planning a follow-up consultation is an important first step in addressing the needs of parents. We therefore recommend GPs to do accordingly, as it fosters an enduring GP–patient relationship, that is beneficial for all conditions, but particularly for patients with FSS. Further research should evaluate whether this framework also holds true for children and explore the views and experiences of GPs about these consultations. Finally, future studies should investigate whether using this framework in daily practice will help GPs manage children with FGID and other FSS clusters and improve the quality of consultation experiences for parents and children.

**Acknowledgements** The authors would like to thank all the respondents who volunteered and took part in this study.

**Contributors** SMA co-designed this study, recruited participants, performed all interviews, performed the initial analyses, interpretation of the data and the first and final draft of the manuscript. SMA is responsible for the overall content of this work (as the guarantor). ING observed all interviews, co-performed the initial and final analysis, took part in the interpretation of the data and revised the manuscript for important intellectual content. DGvT took part in the analysis, interpretation of the data and revised the manuscript for important intellectual content. TF co-designed this study, took part in the analysis, interpretation of the data and revised the manuscript for important intellectual content. MO co-designed this study, took part in the interpretation of the data and revised the manuscript for important intellectual content. MYB co-designed this study, took part in the interpretation of the data and revised the manuscript for important intellectual content. GAH co-designed this study, took part in the interpretation of the data and revised the manuscript for important intellectual content. All authors approved the final version of this manuscript and agree to be accountable for all aspects of the work in ensuring that questions related to the accuracy or integrity of any part of the work are appropriately investigated and resolved. Dr Robert Sykes (www.doctored.org.uk) provided editorial services.

**Funding** This project was supported by ZonMW, Dutch Organization for Health Research and Development (project number 852001930).

**Competing interests** None declared.

**Patient and public involvement** Patients and/or the public were involved in the design, or conduct, or reporting, or dissemination plans of this research. Refer to the Methods section for further details.

**Patient consent for publication** Not applicable.

**Ethics approval** This study involves human participants. This study was approved by the Medical Research Ethics Committee of the University Medical Center Groningen (research register: 202000795). Participants gave informed consent to participate in the study before taking part.

**Provenance and peer review** Not commissioned; externally peer reviewed.

**Data availability statement** Data sharing not applicable as no datasets generated and/or analysed for this study. Due to the confidential and sensitive nature of our data, data will not be made publicly available.

**ORCID iDs**
Sophie M Ansems http://orcid.org/0000-0002-8744-8224
Ilse N Ganzevoort http://orcid.org/0000-0001-9367-7827
Gea A Holtman http://orcid.org/0000-0001-6579-767X

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
