## [Reviewer comments · BMJ Open]

ARTICLE DETAILS

TITLE (PROVISIONAL)	Qualitative study evaluating the expectations and experiences of Dutch parents of children with chronic gastrointestinal symptoms visiting their general practitioner
AUTHORS	Ansems, Sophie; Ganzevoort, Ilse; van Tol, Donald; Fokkema, Tryntsje; Olthof, Marijke; Berger, Marjolein; Holtman, G.

VERSION 1 – REVIEW

REVIEWER	Rajmohan Dharmaraj University of New Mexico, Rajmohan Dharmaraj
REVIEW RETURNED	12-Jan-2023

GENERAL COMMENTS	In this study, Sophie M. Ansems, et al. evaluated the expectations and experiences of parents of children visiting a GP for chronic gastrointestinal symptoms. This is an important topic and very relevant in clinical practice. Authors have done an excellent job conceptualizing framework, and how the GP can improve the quality of patient experiences by recognizing and meeting these needs. I have some suggestions to consider - Authors report that a Dutch general practitioner (GP) will typically see approximately 10 children with chronic gastrointestinal (GI) symptoms each year, of whom about 9 will have a functional gastrointestinal disorder (FGID). In my opinion, majority of GP in other parts of the world would typically see that many patients in a month. Studies have shown recurrent abdominal pain (functional abdominal pain) is very common with a prevalence as high as 20% in general population. Authors may want to explain more about this disparity and if it would explain some of the negative influences.- Did authors notice any difference in outcomes with regards to sex of the parent (mother vs father)?- Did GP's experience and age made a difference in outcomes?
--

REVIEWER	Jason Randall Clinical Outcomes Solutions, Kent, Clinical Outcomes Assessments
REVIEW RETURNED	14-Feb-2023

GENERAL COMMENTS	The manuscript is well-written, and study appears to be well-designed. The study team has access to a niche group of patient participants and have conducted interviews to understand GI symptoms and GP experience. • I am concerned by the fact the primary interviewer deceived her true identity (that she was a physician) from participants in order to elicit information. Was the IRB/ethics board aware that the participants would be deceived, and did they approve this process? If yes please clarify this and add a sentence to this effect. I
---

	completely understand the reason that this was done, to improve rapport, but I would suggest a better approach would have been to have another interviewer conduct the interviews rather than deceive participants.  • Last sentence in data collection doesn't quite make sense and needs more explanation (lines 19-20 page 9) • In the analysis section you refer to "Thematic content analysis", based on the Braun and Clarke reference you cite this should be "thematic analysis", it does not need the word content. Results  • Respondents you have not mentioned how the demog and medical history data was collected, was this patient or clinician reported?  o Table 1 the footnote on education doesn't match the information in the table. The table refers to low medium and high education whilst the footnote uses intermediate instead of medium, please be consistent. o The description of low needs further detail, in parenthesis is has "lower secondary education or lower", what does the second lower refer to? These needs updating • I like table 2 this is very helpful • I have not seen conceptual models presented in this way before, with them being mapped to specific case studies, however, I really like these and the explanation you provide. Nice touch. • In the results, the quotes should be presented in quotation marks. • Nice use of quotes in the results section, these help identify your key points. Discussion  • I agree with your conclusion that there has not been anything published like this before and believe that this may be of interest to other readers • The discussion section needs a bit more work. You only really have 1 paragraph discussing your results which is the first one, I would expand this a bit more. • I would restructure the section you have called strength and weakness, I would remove this as a header, and write this more as a standard discussion of your research. The key points you have raised here are valid and relevant but just need more work to make this more scientific in phrasing and these can also be written far more succinctly.  o I also do not like the use of "we" in the discussion, this should be removed and updated to reflect best practices for scientific manuscript writing. • Line 20-21 on page 19 "Some readers might think a limitation of our study is the 21 lack of data triangulation." Doesn't make sense and is not relevant to your limitations and should be removed. • The key part of your conclusions is highlighting the importance of a good patient and GP relationship. Based on my own experience, as well as reviewing some of the references you have cited, this relationship is important not only in this condition but for any condition. I know this is not the key aim of your study but this seems like an obvious conclusion, yet it is not really touched on although it is loosely alluded to in the final part of your discussion regarding implications. I would like to see this brought out more in your discussion as this seems like an obvious conclusion.
--	---

VERSION 1 – AUTHOR RESPONSE

Reviewer: 1

Dr. Rajmohan Dharmaraj, University of New Mexico

Comments to the Author:

In this study, Sophie M. Ansems, et al. evaluated the expectations and experiences of parents of children visiting a GP for chronic gastrointestinal symptoms. This is an important topic and very relevant in clinical practice. Authors have done an excellent job conceptualizing framework, and how the GP can improve the quality of patient experiences by recognizing and meeting these needs.

We would like to thank you for reviewing our manuscript and providing us with constructive feedback.

I have some suggestions to consider

1. Authors report that a Dutch general practitioner (GP) will typically see approximately 10 children with chronic gastrointestinal (GI) symptoms each year, of whom about 9 will have a functional gastrointestinal disorder (FGID). In my opinion, majority of GP in other parts of the world would typically see that many patients in a month. Studies have shown recurrent abdominal pain (functional abdominal pain) is very common with a prevalence as high as 20% in general population. Authors may want to explain more about this disparity and if it would explain some of the negative influences.

The general population prevalence of recurrent abdominal pain indeed shows a wide range from 0.3 to 19% (Chitkara, Rawat, and Talley 2005) and the pooled prevalence of functional abdominal pain disorders is lowest in Europe (10.5%) compared to other continents (Kortnerink et al. 2015).

Furthermore, it is plausible that the frequency of children consulting for recurrent abdominal pain to Dutch GPs is lower compared to primary caregivers in other health care systems, where pediatricians are responsible for providing primary care. In The Netherlands, children with recurrent abdominal pain are sometimes referred to secondary care, while in other countries, primary care pediatricians manage such cases themselves. Additionally, the use of telephone triage by GP assistants may result in filtering out children with mild symptoms, potentially preventing them from seeking in-person consultation with the GP. This distinction in the healthcare system may have an impact on the consultation rates of children with recurrent abdominal pain.

Given the possibly lower prevalence and consultation frequency of recurrent abdominal pain in The Netherlands, it is possible that Dutch GPs see fewer children with chronic gastrointestinal symptoms than their colleagues outside of Europe and colleagues operating in different health care systems. However, our study, in which only Dutch residents were interviewed, cannot determine if this negatively affected the consultation experiences of children and their parents.

We have added information about the prevalence to our introduction:

'Chronic gastrointestinal symptoms are prevalent in school-aged children, ranging from 0.3 to 19% in Western countries (1). In The Netherlands, a Dutch general practitioner (GP) will typically see approximately 10 children with chronic gastrointestinal (GI) symptoms each year (2,3), of whom about 9 will have a functional gastrointestinal disorder (FGID) (3,4).' (Page 5, Line 2-3)

2. Did authors notice any difference in outcomes with regards to sex of the parent (mother vs father)?

This is an interesting point. Although we purposively recruited more fathers to ensure diversity in our study sample, we did not notice any differences between the views and expectations of mothers and fathers. We have added the following in the Discussion:

'Although we purposively sampled mothers and fathers, we did not find any difference between them.' (Page 20, Line 20-21)

3. Did GP's experience and age made a difference in outcomes?

In the present study, we did not collect information on the GP's experience and age. However, we are currently conducting a qualitative study on the views and experiences of GPs when managing children with chronic gastrointestinal symptoms. In this study, we are collecting the GP's experience and age and will take this demographic information into account while analyzing the interviews.

We now mention in our conclusion that the GP's views and experiences should also be studied:

Further research should evaluate whether this framework also holds true for children. Moreover, it is important to explore the GP's views and experiences of consultations with children with chronic gastrointestinal symptoms. Finally, future studies should investigate whether using this framework in daily practice will help GPs manage children with FGID and other FSS clusters and improve the quality of consultation experiences for parents and children.

Reviewer:

2

Dr. Jason Randall, Clinical Outcomes Solutions, Kent

Comments to the Author:

The manuscript is well-written, and study appears to be well-designed. The study team has access to a niche group of patient participants and have conducted interviews to understand GI symptoms and GP experience.

We would like to thank you for reviewing our manuscript and providing us with constructive feedback.

4. I am concerned by the fact the primary interviewer deceived her true identity (that she was a physician) from participants in order to elicit information. Was the IRB/ethics board aware that the participants would be deceived, and did they approve this process? If yes please clarify this and add a sentence to this effect. I completely understand the reason that this was done, to improve rapport, but I would suggest a better approach would have been to have another interviewer conduct the interviews rather than deceive participants.

We would like to thank the reviewer for bringing up this interesting ethical and methodological conflict. However, we do not agree with the reviewer that we deceived the participants. Every individual has multiple social roles and identities and so do researchers. In qualitative research it is common and good practice to only disclose the roles/identities of the interviewer that are important for the current research question in the current context. By not telling the parents and/or children that the interviewer was a physician besides her role as researcher/interviewer we gave them room to speak freely about their GP and about doctors in general and made them feel safe to also share their negative experiences. Additionally, if the interviewer would have told the participants that she was a physician, this could have created false expectations. The interview was not a doctor's consultation and was not scheduled to answer questions about their child's health.

In summary, we believe this was good research practice and no deception. We have added a reference to an article describing the role of the researcher in qualitative studies (Collins and Stockton 2022) to support this (page 8, line 9).

We specifically would like to point out the following section from this article:

'Goffman (1959) conceptualized masking to be part of the process of how we conceive ourselves, what we are seeking to be, and ultimately can be incorporated as an integral part of our personalities. If a person does not believe in or see themselves in the role they are playing, they could be engaging in some form of deception (even self-deception) or misrepresentation. In the case of research and the evolving researcher, the function of deception is an ethical conflict. Special permission is typically

required to use deception in a study with human subjects. However, the masking of a portion of self and only divulging enough to play the role of interviewer, is not deception—until the researcher says something that they do not actually believe. At that point the interviewing role becomes a breach of self and could be considered deception.’

We have also added a reference to the book ‘Qualitative Research Methods’ (Hennink, Hutter, and Bailey 2020) (page 8, line 9). We refer to the following section (page 127) about the positionality of the researcher:

‘Your positionality, that is, how you present yourself in terms of your role or title, can also influence the interviewee. Positionality refers to the power relations between interviewer and the interviewee. For example, it will make a huge difference whether you introduce yourself in an interview as university professor or as a researcher who is interested in the life of people. From the very first moment, this will determine the power relationship between the interviewer and the interviewee, and therefore the information you can collect. Furthermore, you need to reflect on whether it would be possible for a man to interview a woman and vice versa. During the interviews, you become aware of the power relations between you and the interviewee, and how it may influence what the interviewee shares about their life and experiences. Does this mean that a researcher or interview can select any role to play while conducting in-depth interviews? You cannot change certain personal characteristics, such as being a woman or man, having a white, brown or black skin or your physical features. However, there are characteristics that one can decide to highlight or not. Ethical issues are of course important here: it would not be ethical, for example, for a childless female researcher conducting research on reproduction to pretend that she has children herself. Reflecting on both your subjectivity and positionality is important before, during and after conducting in-depth interviews. Before and during interviews you need to be aware of, and reflect on, how you present yourself to the study population and how you can establish rapport.’

5. Last sentence in data collection doesn’t quite make sense and needs more explanation lines 19-20 page 9

We made the following changes (Page 8, line 19-20):

‘Parents and/or children received an interview synopsis ~~to perform a member check~~ and checked whether the interpretation of the interview was correct (member check 1).’

6. In the analysis section you refer to “Thematic content analysis”, based on the Braun and Clarke reference you cite this should be “thematic analysis”, it does not need the word content.

Thank you for your attentive reading. We have removed ‘content’.

Results

7. Respondents you have not mentioned how the demog and medical history data was collected, was this patient or clinician reported?

This was both patient and clinician reported. We have added this information to Table 1.

Table 1. Parents’ and children’s characteristics (n=15)

Characteristic	Number
Gender child, male ^a	7

Age child, years ^a	
4 -7	5
8 - 11	8
12 - 15	1
16 - 17	1
Urban area of residence ^a	
	5
Country of birth of both parents ^b	
The Netherlands	15
Educational level mother ^{b, c}	
Low	1
Medium-Intermediate	6
High	8
Educational level father ^{b, c}	
Low	2
Medium-Intermediate	6
High	7
Paid employment ^b	
Both parents	14
One parent	1
Marital status parents ^b	
Married / living together	11
Divorced	4
Present at interview ^b	
Child	1
Child and mother	1
Mother	8
Father	2
Both parents	3

Diagnosis after 6 months ^a

Functional constipation	9
Irritable bowel syndrome	2
Functional constipation and functional abdominal pain	2
Functional abdominal pain	2

^a Clinician reported

^b Patient reported

^c Educational level was divided in low (primary and lower secondary education or lower), intermediate (secondary (vocational) education) and high (Bachelor's degree or higher) (Centraal Bureau voor Statistiek (CBS) n.d.; UNESCO Institute for Statistics 2011).

8. Table 1 the footnote on education doesn't match the information in the table. The table refers to low medium and high education whilst the footnote uses intermediate instead of medium, please be consistent.

Thank you for your careful reading, we have changed medium to intermediate in the Table (see answer to comment 7).

o The description of low needs further detail, in parenthesis is has "lower secondary education or lower", what does the second lower refer to? These needs updating

We understand this was confusing. The second lower referred to primary education (primary school). We have changed this in the table's footnote (see answer to comment 7)

9. I like table 2 this is very helpful

10. I have not seen conceptual models presented in this way before, with them being mapped to specific case studies, however, I really like these and the explanation you provide. Nice touch.

Thank you for your positive feedback.

11. In the results, the quotes should be presented in quotation marks.

We have added quotation marks to the quotes.

12. Nice use of quotes in the results section, these help identify your key points.

Thank you.

Discussion

13. I agree with your conclusion that there has not been anything published like this before and believe that this may be of interest to other readers.

Thank you.

14. The discussion section needs a bit more work. You only really have 1 paragraph discussing you results which is the first one, I would expand this a bit more.

We have expanded our principal findings (first paragraph) with the following sentence (which we moved from the conclusion):

Within this framework, specific needs of parents influenced the themes and their interrelationships: for the GP to acknowledge the disease burden, to search for an explanation together with the GP and to feel reassured. GPs should be aware that when they recognize and respond to these needs, they can positively influence the consultation experience of parents.

To improve the flow of the discussion we have removed the headers and re-organized it. Right after the first paragraph, we now discuss our findings in relation to existing literature. We then discuss our strengths and limitations. We end with the meaning of our study, which now focuses more on the implications for clinicians and future research. For the detailed changes, please see the manuscript (page 18-page 22).

15. I would restructure the section you have called strength and weakness, I would remove this as a header, and write this more as a standard discussion of your research. The key points you have raised here are valid and relevant but just need more work to make this more scientific in phrasing and these can also be written far more succinctly.

Please see our answer to comment 14. We tried to re-write our discussion more succinctly.

16. I also do not like the use of “we” in the discussion, this should be removed and updated to reflect best practices for scientific manuscript writing.

Please see our answer to comment 14. We have re-written our discussion and now limit the use of ‘we’.

17. Line 20-21 on page 19 “Some readers might think a limitation of our study is the lack of data triangulation.” Doesn’t make sense and is not relevant to your limitations and should be removed.

We removed this sentence (please see page 21).

18. The key part of your conclusions is highlighting the importance of a good patient and GP relationship. Based on my own experience, as well as reviewing some of the references you have cited, this relationship is important not only in this condition but for any condition. I know this is not the key aim of your study but this seems like an obvious conclusion, yet it is not really touched on although it is loosely alluded to in the final part of your discussion regarding implications. I would like to see this brought out more in your discussion as this seems like an obvious conclusion.

We agree that a good GP-patient relationship is a core principle of general practice medicine, and is therefore important in any given condition. We have added the following to our conclusion to emphasize this:

Although in our study the appropriate GP response was situation specific, planning a follow-up consultation is an important first step in addressing the needs of parents. We therefore recommend GPs to do accordingly, as it fosters an enduring GP–patient relationship, that is beneficial for all conditions, but particularly for patients with FSS.

VERSION 2 – REVIEW

REVIEWER	Rajmohan Dharmaraj University of New Mexico, Rajmohan Dharmaraj
----------	--

REVIEW RETURNED	25-Apr-2023
GENERAL COMMENTS	The authors incorporated my suggestions in the manuscript. In my opinion, the revised manuscript can be published if other reviewers concerns are addressed.
REVIEWER	Jason Randall Clinical Outcomes Solutions, Kent, Clinical Outcomes Assessments
REVIEW RETURNED	20-Apr-2023
GENERAL COMMENTS	The edits made have helped improve the readability of the manuscript.